# TEMPORAL GRAPH REWIRING WITH EXPANDER GRAPHS

## ABSTRACT

Evolving relations in real-world networks are often modelled by temporal graphs. Temporal Graph Neural Networks (TGNNs) emerged to model evolutionary behaviour of such graphs by leveraging the message passing primitive at the core of Graph Neural Networks (GNNs). It is well-known that GNNs are vulnerable to several issues directly related to the input graph topology, such as under-reaching and over-squashing—we argue that these issues can often get *exacerbated* in temporal graphs, particularly as the result of stale nodes and edges. While graph rewiring techniques have seen frequent usage in GNNs to make the graph topology more favourable for message passing, they have not seen any mainstream usage on TGNNs. In this work, we propose Temporal Graph Rewiring (TGR), the first approach for graph rewiring on temporal graphs, to the best of our knowledge. TGR constructs message passing highways between temporally distant nodes in a continuous-time dynamic graph by utilizing expander graph propagation, a prominent framework used for graph rewiring on static graphs which makes minimal assumptions on the underlying graph structure. On the challenging TGB benchmark, TGR achieves state-of-the-art results on `tgbl-review`, `tgbl-coin`, `tgbl-comment` and `tgbl-flight` datasets at the time of writing. For `tgbl-review`, TGR has 50.5% improvement in MRR over the base TGN model and 22.2% improvement over the base TNCN model. The significant improvement over base models demonstrates clear benefits of temporal graph rewiring.

## 1 INTRODUCTION

Graph representation learning (Hamilton, 2020) aims to learn node representations on graph structured data to solve tasks such as node property prediction (Hamilton et al., 2017; Kipf & Welling, 2016), link prediction (Ying et al., 2018; Zitnik et al., 2018) and graph property prediction (Gilmer et al., 2017). Graph neural networks (GNNs) (Kipf & Welling, 2016; Veličković et al., 2017; Xu et al., 2018) capture relationships between nodes following the message passing paradigm (Gilmer et al., 2017). GNNs have been successfully applied to model many real-world networks such as biological networks (Johnson et al., 2023; Zitnik et al., 2018) and social networks (Ying et al., 2018).

**Graph rewiring**. GNNs operate through a message passing mechanism which aggregates information over a node's *direct* neighbourhood at each GNN layer. It is by now well known that such an approach exposes GNNs to several vulnerabilities that are inherently tied to the input graph's topology, such as *under-reaching* (Barceló et al., 2020) and *over-squashing* (Alon & Yahav, 2021; Di Giovanni et al., 2024). In *static* GNNs, such issues are frequently addressed using **graph rewiring**, where the input graph is altered such that it connects distant nodes. There are several methods applied to rewire static GNNs such as diffusion-based graph rewiring models (Gasteiger et al., 2019) or reducing negative Ricci curvature (Topping et al., 2021), leading to a boost in performance.

**Temporal graph learning (TGL)**. Temporal graph learning is an emerging field which aims to study evolving relations in dynamic real world networks (Kazemi et al., 2020). These evolving relations are modelled by *temporal graphs*, wherein nodes and edges can be inserted or deleted over time. Accordingly, a more generic class of *temporal* graph neural networks (TGNNs) (Longa et al., 2023) are designed to capture the evolution of such graphs by introducing novel model components such as temporal memory (Kazemi et al., 2020) and time-encoding (Xu et al., 2020).

TGNNs are underpinned by a static GNN message passing mechanism, and this makes them vulnerable to the same kinds of under-reaching and over-squashing effects faced by static GNNs. In this work we will show that the addition of a temporal dimension introduces an additional *hierarchical* flow of information along the axis of time, which provably *exacerbates* these vulnerabilities and makes it even harder for nodes to meaningfully communicate. A natural question arises: *Can we alleviate these issues and improve TGNN performance by introducing **temporal** graph rewiring?*

**Benefits of temporal graph rewiring**. Graph rewiring fundamentally embraces the *dynamic* nature of real-world data. It is quite rare that any particular input graph perfectly elucidates the required information exchange for a given task, and this might be even more relevant in a temporal graph, where observable links are constantly subject to change. Additionally, temporal graph rewiring can be seen as a way to resolve *memory staleness* problems in temporal GNNs (Kazemi et al., 2020). By memory staleness we refer to a process occurring in the temporal memory of TGNNs:

The temporal memory is only updated if a node of interest is observed to interact with another node in a graph, and this causes inactive nodes' states to become *stale*. This poses significant issues in many real world dynamic graphs; e.g., in social networks, if a user is inactive for a period of time, they lose connections to their active friends, even if they still interact with them outside of the social network. We posit that the additional connections created by temporal graph rewiring will alleviate this effect by allowing information to consistently flow between users that might otherwise be inactive for a longer period of time.

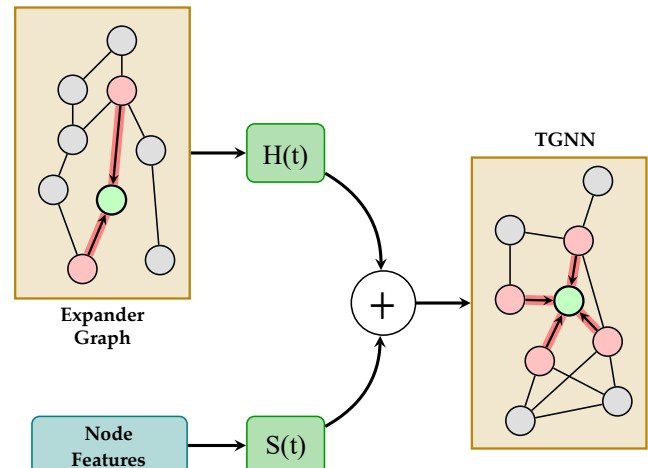

Figure 1: Expander graph emeddings are mixed with input node features to rewire the base TGNN model in TGR.

In this work, we propose the **TGR** framework for *Temporal Graph Rewiring* with expander graphs. We leverage recent work by Deac et al. (2022) on expander graph propagation in combination with a TGNN base model to ensure global message passing between *temporally* distant nodes. It is shown that expander graphs satisfy four desirable criteria for graph rewiring: 1) the ability to efficiently propagate information globally within the input graph, 2) relieving bottlenecks and over-squashing, 3) subquadratic space and time complexity, and 4) no additional pre-processing of the input graph. As shown in Figure 1, TGR operates by utilizing expander graphs to form message passing highways between temporally distant nodes, alleviating issues such as under-reaching and memory staleness.

Our main contributions can be summarised as follows:

- **The *theoretical* motivation for temporal graph rewiring.** We shed light on the fact that important challenges in static graphs, such as *under-reaching* (Barceló et al., 2020), are made more challenging when graphs gain a temporal axis. This directly motivates the need for rewiring over temporal graphs.

- **The *first* rewiring method for temporal graphs.** To respond to this need, we propose *Temporal Graph Rewiring* or TGR. To the best of our knowledge, TGR is the first approach which applies graph rewiring techniques on temporal graphs. TGR leverages expander graph propagation (Deac et al., 2022) to rewire a base temporal graph neural network such as TGN (Rossi et al., 2020), while operating with minimal additional overhead. Further, TGR is **agnostic** to the choice of base temporal graph learning model; we demonstrate this by evaluating TGR over two widely used TGNNs (Rossi et al., 2020; Zhang et al., 2024).

- **The *state-of-the-art* in temporal graph representation learning.** We test TGR on temporal link prediction tasks within the Temporal Graph Benchmark (Huang et al., 2023, TGB). In Section 5, we show that TGR consistently outperforms other baseline TGNNs, while setting the new

state-of-the art performance on four datasets (`tgbl-review`, `tgbl-coin`, `tgbl-comment` and `tgbl-flight`), often by a significantly wide margin. This result indicates that temporal graph rewiring indeed unlocks communication between node pairs that were relevant and critical for diverse real-world temporal graph tasks.

**Reproducibility.** Our code is available on 4open.science and will be made public.

## 2 RELATED WORK

**Temporal graph neural networks**. Graph-based TGNNs (Yu et al., 2023; Cong et al., 2023; Xu et al., 2020; Wang et al., 2022) leverage structural features of the input graph for learning temporal graph representations. Although these architectures achieve state-of-the-art performance on smaller TGB datasets (Huang et al., 2023), they rely on heavy processing of structural features, resulting in limited opportunities for scaling. Memory-based TGNNs (Rossi et al., 2020; Trivedi et al., 2019; Johnson et al., 2023) use temporal memory to retain historical node information by utilizing additional model components such as recurrent networks (Rossi et al., 2020) or two-time scale deep temporal point processes (Trivedi et al., 2019) to dynamically keep track of node interactions as they occur. Recently, Zhang et al. (2024) extend Neural Common Neighbor (NCN) to the temporal domain, leveraging pairwise representations and link-specific features. Memory-based architectures are successful in processing larger datasets, however they lack the ability to observe long-range dependencies between nodes in dynamic graphs. In contrast, our method demonstrates ability to both scale well with increase of dataset size and observe long-range dependencies between temporal data points, due to its usage of expander graphs, which require no additional dynamic pre-processing.

**Static graph rewiring**. Graph rewiring has been employed to address bottlenecks and over-squashing in static GNNs, leading to an impressive boost in performance. For instance, diffusion-based graph rewiring (Gasteiger et al., 2019) diffuses additional edges in the graph with use of kernels such as PageRank (Page et al., 1999). However, as stated by Topping et al. (2021), such models generally fail to reduce bottlenecks in the input graph. In contrast, Topping et al. (2021) propose to modify a portion of edges guided by their Ricci curvature. However, this method incurs high pre-processing cost and it is sub-optimal for analyzing large graphs, such as continuous-time dynamic graphs, which are the focus of our paper. Deac et al. (2022) introduce *expander graph propagation* (EGP), which, unlike prior rewiring methods, requires no dedicated preprocessing: EGP propagates information over an expander graph which is *independent* from the input graph. EGP thus minimises computational overhead while exhibiting favourable topological properties for message passing. This makes EGP our preferred method of choice for TGR, as its rewiring side-steps the added temporal complexity of dynamic graphs.

## 3 THEORETICAL BACKGROUND AND MOTIVATION

This section will provide a brief formal treatment of temporal graphs, describe the TGNN class of models over them, and motivate temporal graph rewiring by analysing how TGNNs are exposed to vulnerabilities in the input graph topology, especially in comparison to static graphs.

### 3.1 TEMPORAL GRAPHS

Temporal graphs are characterised by a topology that *evolves* through time, thus making them a suitable abstraction for modelling complex dynamic networks. We provide a formal definition next.

**Continuous-time dynamic graphs**. In this work, we focus on continuous-time dynamic graphs (CTDGs), that are used to model events occurring at *any* point in time, thus offering a versatile framework for representing real-world networks.

**Definition 1** (Continuous-time dynamic graph). *Let $T \in \mathbb{R}^+$ denote the current time. A continuous-time dynamic graph (CTDG) is a multi-graph $G_T = (V(T), E(T), \mathbf{X}(T))$ comprising a set of all nodes that appeared in this time $V(T)$, and a stream of chronologically*

*sorted edges $(u, v, t) \in E(T)$ between nodes $u, v \in V(T)$ that occurred at time $t \in [0, T)$. $\mathbf{X}(T) \in \mathbb{R}^{|V(T)| \times k}$ represents the $k$-dimensional input features of all observed nodes.*

Definition 1 can be specialised to specify a *snapshot* of a CTDG at time $\tau \in [0, T)$, denoted by $G_{\leq \tau} = (V_{\leq \tau}, E_{\leq \tau}, \mathbf{X}(\tau))$, which contains a collection of nodes and edges that were observed during the time interval $[0, \tau]$. Over this period of time, nodes need not all be active simultaneously, and the sequence of timesteps at which nodes are observed significantly influences the dynamics in CTDGs. We denote $t_u \in [0, \tau]$ as the last activation time of a node $u \in V_{\leq \tau}$, i.e., the time at which node $u$ last appears in the CTDG's snapshot up to $\tau$.

For a pair of nodes $u, v \in V_{\leq \tau}$, we use $E_{\leq \tau}^{(u,v)} \subseteq E_{\leq \tau}$ to represent the set of all historical interactions betweeen $u$ and $v$ during the observed time period $[0, \tau]$. We also denote their associated node features as $\mathbf{x}_u^{(\tau)}, \mathbf{x}_v^{(\tau)} \in \mathbb{R}^k$. Lastly, we may also consider the *open* snapshot of all events prior to time $\tau$, $G_{< \tau}$, considering all events that were observed within $[0, \tau)$.

**Temporal link prediction tasks**. We will be particularly interested in tasks requiring the prediction of existence or absence of links in continuous-time dynamic graphs. Such tasks have significant implications for real-world recommendation systems, especially in social networks. To formalise this, we will assume that predicting links in a CTDG consists of computing temporal embeddings of its nodes, and then predicting a probability of an edge's existence between a pair of nodes given those two nodes' embeddings. Note this is a simplified framework—many TGNNs may also compute explicit temporal edge embeddings, for example.

Formally, we assume that a temporal link prediction task at time $\tau$ focuses on predicting existence of an edge between a pair of nodes $u, v \in V_{\leq \tau}$ in a CTDG snapshot $G_{\leq \tau}$. Hence, a temporal link prediction task can be formulated as deciding whether $(u, v, \tau) \in E_{\leq \tau}^{(u,v)}$, as a function of $G_{< \tau}$.

## 3.2 TEMPORAL GRAPH NEURAL NETWORKS

In order to better understand the vulnerabilities of temporal graph neural networks (TGNNs) with respect to the temporal graph topology, we need to briefly formulate how they aggregate information along the input CTDG. For the purposes of making our analysis more elegant, we will make two simplifying assumptions:

- Incoming edges are processed one at a time, and no two edges occur at the same time, i.e. for any $u, v, w, z \in V_{\leq \tau}$ such that $u \neq w \lor v \neq z$, $(u, v, \tau) \in E_{\leq \tau} \implies (w, z, \tau) \notin E_{\leq \tau}$. We also assume that there are no additional edge features provided.
- All nodes in the CTDG are observed from the beginning ($V_{\leq \tau} = V_{\leq 0}$) and their input features are never updated ($\mathbf{X}(\tau) = \mathbf{X}(0)$).

Our exposition follows a more abstractified variant of the TGN model (Rossi et al., 2020), which maintains a temporal memory $\mathbf{S}(\tau) \in \mathbb{R}^{|V_{\leq \tau}| \times k}$ which summarises all the knowledge about a node $u \in V_{\leq \tau}$ up to time $\tau$ within a memory vector $\mathbf{s}_u^{(\tau)} \in \mathbb{R}^m$.

Initially, the memory vectors are set as $\mathbf{s}_u^{(0)} = \chi(\mathbf{x}_u^{(0)})$ for a learnable function $\chi : \mathbb{R}^k \to \mathbb{R}^m$. Upon encountering a new event $(u, v, \tau)$, a TGN updates the memory vectors of $u$ and $v$ accordingly:

$$\mathbf{s}_u^{(\tau)} = \phi_s\left(\mathbf{s}_u^{(\tau-\epsilon)}, \mathbf{s}_v^{(\tau-\epsilon)}\right) \qquad \mathbf{s}_v^{(\tau)} = \phi_d\left(\mathbf{s}_v^{(\tau-\epsilon)}, \mathbf{s}_u^{(\tau-\epsilon)}\right)$$

where $\epsilon > 0$ is a suitably chosen constant such that no edges are observed in the time interval $[\tau - \epsilon, \tau)$, and $\phi_s, \phi_d : \mathbb{R}^m \times \mathbb{R}^m \to \mathbb{R}^m$ are learnable *update functions*.

The memory vectors $\mathbf{S}(\tau)$ can then be freely leveraged to answer temporal link prediction queries at time $\tau + \epsilon$ on demand. TGNs perform this by computing temporal node embeddings $\mathbf{Z}(\tau) \in \mathbb{R}^l$ as a function of each node's temporal neighbourhood, $\mathcal{N}_{\leq \tau}^{(u)} = \{v \mid (u, v, \tau') \in E_{\leq \tau} \lor (v, u, \tau') \in E_{\leq \tau}\}$. Abstractly:

$$\mathbf{z}_u^{(\tau)} = \bigoplus_{v \in \mathcal{N}_{\leq \tau}^{(u)}} \psi\left(\mathbf{s}_u^{(\tau)}, \mathbf{s}_v^{(\tau)}, \mathbf{x}_u^{(0)}, \mathbf{x}_v^{(0)}\right)$$

where $\psi : \mathbb{R}^m \times \mathbb{R}^m \times \mathbb{R}^k \times \mathbb{R}^k \to \mathbb{R}^l$ is a learnable *message function*, and $\bigoplus : \text{bag}(\mathbb{R}^l) \to \mathbb{R}^l$ is a permutation-invariant aggregation function, such as sum, average or max.

Finally, the computed embeddings are used to compute the relevant probabilities for link prediction:

$$\mathbb{P}((u, v, \tau + \epsilon) \in E_{\leq \tau + \epsilon}) \propto \kappa \left( \mathbf{z}_u^{(\tau)}, \mathbf{z}_v^{(\tau)} \right)$$

where $\kappa : \mathbb{R}^l \times \mathbb{R}^l \to \mathbb{R}$ is a learnable logit function. These logits can be then optimised towards correct values using gradient descent, as the architecture is fully differentiable.

### 3.3 TEMPORAL UNDER-REACHING

The design of TGNs, particularly its *local* memory updates, represents an elegant tradeoff between expressive power and favourable computational complexity. However, as we will show, this design leaves TGNs vulnerable to more severe forms of the *under-reaching* effect (Barceló et al., 2020). Under-reaching concerns itself with the questions of the form: is node $u$ able to make any local decisions in a way that depends on features of another node $v$? Or, equivalently, will features of node $v$ *mix* into the embeddings of node $u$?

For $k$-layer GNNs over *static* graphs $G = (V, E)$, under-reaching is simple to define: it occurs between any two nodes $u, v \in V$ for which $k < d_G(u, v)$, where $d_G(u, v)$ is the shortest path length between $u$ and $v$ in $G$. Clearly, it will not be possible for the information from node $v$ to mix into node $u$'s embeddings in this case, as each GNN layer mixes information that is one hop further.

What happens in *temporal* graphs $G_{\leq \tau}$, for our previously studied TGNN framework?

First, note that, since the computed temporal node embeddings $\mathbf{z}_u^{(\tau)}$ are used only for answering link prediction queries and they are *not* committed back into memory, we can focus on studying mixing over temporal memory vectors $\mathbf{s}_u^{(\tau)}$ only, and handle the embedding mixing as a follow-up case.

For now, we will assume that the input features of node $u$ are *temporally mixed* into node $v$'s memory at time $\tau$ if $\mathbf{s}_v^{(\tau)}$ depends on information from $\mathbf{x}_u^{(0)}$. Since a node's memory vector is only updated when that node is observed within an edge, we can formally define temporal mixing and under-reaching by tracking paths by which this information travels:

**Definition 2** (Temporal under-reaching). *Let $G_{\leq \tau}$ be a continuous-time dynamic graph snapshot, and let $a \overset{t}{\longleftrightarrow} b$ mean that either $(a, b, t) \in E_{\leq \tau}$ or $(b, a, t) \in E_{\leq \tau}$. Then, the features of node $u$ are said to be **temporally mixed** into node $v$'s memory at time $\tau$ by a TGNN, if there exists a sequence of nodes $u \overset{t_0}{\longleftrightarrow} w_1 \overset{t_1}{\longleftrightarrow} \ldots \overset{t_{n-1}}{\longleftrightarrow} w_n \overset{t_n}{\longleftrightarrow} v$ such that $t_0 < t_1 < \cdots < t_n$. If no such sequence exists, **temporal under-reaching** occurs from node $u$ to node $v$ at time $\tau$, and node $v$'s memory vector, $\mathbf{s}_v^{(\tau)}$, cannot depend on node $u$'s input features, $\mathbf{x}_u^{(0)}$.*

While temporal under-reaching at time $\tau$ may appear related to static under-reaching of $G_{\leq \tau}$, it occurs *substantially* more frequently, as we can show by the following proposition:

**Proposition 1** (Temporal under-reaching is more severe than static under-reaching). *There exists a continuous-time dynamic graph snapshot $G_{\leq \tau}$ which exhibits temporal under-reaching from node $u$ to node $v$ at time $\tau$, while running a static GNN over all edges in $G_{\leq \tau}$ at once for the same number of layers would not exhibit under-reaching.*

To show this proposition is true, we can consider a simple "path temporal graph" where the only edges are $(u, w_1, t_0), (w_1, w_2, t_1), \ldots, (w_n, v, t_n)$. Clearly, a static GNN over this temporal graph, ran over $n + 1$ layers, does not exhibit under-reaching between $u$ and $v$. However, $u$ *temporally* under-reaches $v$ in almost all configurations—the only exception is if $t_0 < t_1 < \cdots < t_n$.

We can use this same graph to illustrate another relevant result, which holds in temporal under-reaching but not when the graph is static:

**Proposition 2** (Temporal under-reaching is asymmetric). *There exists a continuous-time dynamic graph snapshot $G_{\leq \tau}$ which exhibits temporal under-reaching from node $u$ to node $v$ at time $\tau$, while not exhibiting temporal under-reaching from node $v$ to node $u$ at time $\tau$.*

In the path temporal graph, if node $u$ does not temporally under-reach into node $v$, it *must* hold that node $v$ temporally under-reaches into $u$, settling the proposition.

### 3.4 FURTHER COMPLICATIONS

Note that we demonstrated temporal under-reaching in the "most optimistic possible" case, wherein we're starting with a static initial set of features ($\mathbf{X}(0)$) and merely aiming to spread it using the temporal edges. In practice, there are several aspects of TGNNs' implementation that further exacerbate the under-reaching effect. In all of the below examples, we track a possible mixing path $u \overset{t_0}{\longleftrightarrow} w_1 \overset{t_1}{\longleftrightarrow} \ldots \overset{t_{n-1}}{\longleftrightarrow} w_n \overset{t_n}{\longleftrightarrow} v$.

**Temporal batching**. In practice, temporal edges are seldom processed one-at-a-time; for efficiency reasons, they are divided into temporal batches, where groups of edges happening in the same time interval are processed together. Concretely, we break up our interval into pieces $b_0 = [0, \tau_0), b_1 = [\tau_0, \tau_1), \ldots$, and we denote by $b(t) \in \mathbb{N}$ the batch identifier in which an edge at time $t$ would be processed. Since all edges in a batch propagate simultaneously, this means they cannot feature on the same mixing path. Hence the constraint on our tracked path becomes $b(t_0) < b(t_1) < \cdots < b(t_n)$ for temporal mixing to occur, and this is a stricter condition than $t_0 < t_1 < \cdots < t_n$.

**Dynamic node/edge updates**. It is quite common that new information may enter the temporal graph beyond edge addition, either simply as novel node or edge features to be taken into account, or even entirely new nodes entering the network. In this case, we are attempting to analyse a more generic form of temporal mixing: "If a node/edge is updated at time $t$, does this information reach another node in time for link prediction at time $\tau$?". This translates to a condition $t \leq t_0 < t_1 < \cdots < t_n$, as any edges happening before the information came in cannot be used to propagate it through the memory. Once again, this is a stricter condition than the one we started from, and has particular potential to create nodes with *stale memory*.

### 3.5 WHAT ABOUT THE TEMPORAL NODE EMBEDDINGS?

Our discussion of temporal under-reaching so far focused exclusively on the temporal memory $\mathbf{S}(\tau)$. This has been done for good reason: the computation of temporal node embeddings $\mathbf{Z}(\tau)$ is effectively done over a static version of $G_{\leq \tau}$, and if we allow the temporal neighbourhood $\mathcal{N}_{\leq \tau}^{(u)}$ to be wide enough, this module could eliminate any under-reaching issues.

However, recall that the embedding model needs to be executed *every time* a new temporal link prediction query is raised. Hence, there is an inherent tradeoff: making the embedding module too deep may easily lead to untenable computational complexity as the dynamic graphs get larger. Indeed, in most practical implementations the diameter of the embedding neighbourhood is no more than two hops—which only reduces our path constraint for temporal mixing by two steps.

There is another important issue which may make it challenging to use the embedding module—that is **node/edge deletion**. Deleting a node or edge before time $\tau$ also deletes it from the static graph of $G_{\leq \tau}$, and may lead to full discontinuities in it. In such cases, the embedding module will be provably unable to mix certain nodes together. Rather than overinflating the embedding module, in TGR we seek a lightweight manner to propagate information more widely across the graph, without relying too much on the provided temporal edges, while also making sure that the propagated information is committed back to memory.

## 4 TEMPORAL GRAPH REWIRING (TGR)

This section presents the Temporal Graph Rewiring (TGR) framework. The main components of TGR include *memory mixing* that uses expander graphs to induce mixing between under-reaching nodes and *expander embeddings* that translate this information to the base TGNN model. TGR

builds upon the expander graph propagation framework (Deac et al., 2022), known for alleviating over-squashing and under-reaching in static graphs, and adapt it for temporal graph learning. As illustrated in Figure 2, TGR enhances input node features of observed nodes with expander embeddings that are computed through *memory mixing*. Memory mixing facilitates information exchange between disconnected and distant nodes which are vulnerable to issues such as *under-reaching* and *memory staleness*. By utilizing the expander embeddings to enhance input node features for TGNNs, TGR provides additional information about potentially unreachable nodes to the base TGNN model. As CTDGs are parsed as chronologically ordered batches of edges, TGR computes expander embeddings at the prior batch and includes them for the current batch, naturally integrating into the workflow of a TGNN model.

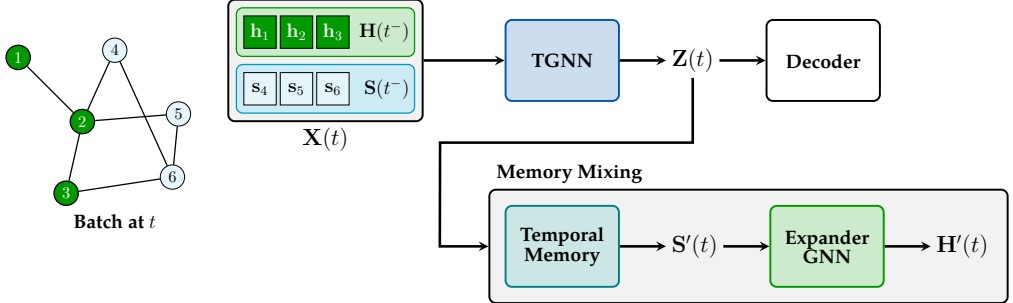

Figure 2: Green nodes are observed nodes and blue nodes are new nodes. Input node features $\mathbf{X}(t)$ are constructed by concatenating expander embeddings $\mathbf{H}(t^-)$ (for observed nodes), and node states $\mathbf{S}(t^-)$ (for new nodes). After computing temporal embeddings $\mathbf{Z}(t)$, temporal memory is updated and *mixed* with other node states in temporal memory to compute expander embeddings.

### 4.1 EXPANDER GRAPH PROPERTIES

Expander graphs are a fundamentally sparse family of graphs ($|E| = O(n)$) that exhibit favourable properties beneficial for relieving over-squashing and under-reaching. This section provides insight into their topological properties: *easy construction*, *no bottlenecks*, and *optimized commute time*.

**Efficiently Precomputable**. Expander graphs are constructed through use of group operators and act as an *independent* graph topology over which input graph is rewired. Our method considers *Cayley graph* expander family, which are constructed through a use of the *special linear group* $\mathrm{SL}(2, \mathbb{Z}_n)$ as a generating set. Further details on constructing Cayley graphs can be found in work by Kowalski (2019); Selberg (1965); Davidoff et al. (2003).

**No Bottlenecks**. The spectral gap of a graphis closely relate to its connectivity and the existence of bottlenecks within its structure. It is known that the first non-zero eigenvalue of the graph Laplacian matrix is lower-bounded by a positive constant $\epsilon > 0$ in expander graphs (Alon, 1986) and $\epsilon = \frac{3}{16}$ for Cayley graph (Selberg, 1965). Given their tree-like structure (Deac et al., 2022), this concludes that expander graphs and particularly Cayley graphs have *high connectivity* and *no bottlenecks*, following Cheeger inequality.

**Optimized Commute Time**. Recent work by Di Giovanni et al. (2024), formally validates that such approach is effective in relieving over-squashing by decreasing overall effective resistance or commute time (Chandra et al., 1989). In fact, commute time is shown to be closely aligned with over-squashing, stating that commute time in expander graphs grows linearly with the number of edges.

### 4.2 EXPANDER GRAPH PROPAGATION FOR GRAPH REWIRING

We briefly discuss key details behind expander graph propagation framework in static setting. Given a set of input node features $\mathbf{X}^{n \times d}$, where $n$ denotes the number of nodes and $d$ represents the feature vector size, expander graph propagation operates by alternating GNN layers and propagating information stored in nodes over the adjacency matrices of both the input graph $\mathbf{A}$ and a Cayley

graph $\mathbf{A}^{\mathrm{Cay}}$ to compute expander node embeddings $\mathbf{H}$ as follows:

$$\mathbf{H} = \mathrm{GNN}(\mathrm{GNN}(\mathbf{X}, \mathbf{A}), \mathbf{A}^{\mathrm{Cay}}).$$

Given this construction, $\mathrm{GNN}(\cdot, \cdot)$ can be any classical GNN layer, such as graph attentional network (GAT) (Veličković et al., 2017):

$$\mathbf{h}_i = \Big\|_{k=1}^{K} \sigma \left( \sum_{j \in \mathcal{N}_i} \alpha_{ij}^k \mathbf{W}^k \mathbf{x}_j \right).$$

In the multi-head attention layer equation above, $i$ denotes a node of interest and $\mathcal{N}_i$ is its 1-hop direct neighbourhood. The attention weights $\alpha_{ij}^k$ correspond to the $k$-th attention head and $\mathbf{W}^k$ represents the input linear transformation's weight matrix.

### 4.3 Expander Graph Propagation on Temporal Graphs

This set-up utilizes underlying expander graph by virtually creating temporal paths that connect under-reaching nodes, independently of their prior interactions in a CTDG. Observed nodes are stored in a node bank module, which dynamically updates its content at every temporal batch. Resulting expander embeddings are used to enhance input node features of observed nodes in the node bank and feed exchanged information into the base TGNN model as dynamic input node features.

By providing dynamic node features, TGR remains agnostic to the underlying TGNN architecture, allowing easy integration with various TGNN models without altering their layer construction. This set-up differs from static graph rewiring methods that modify individual GNN layers within the model architecture by adding a rewiring mask to alter input graph topology.

In this construction, a large expander graph is pre-computed at the beginning of training to match the size of temporal memory. We construct input features $\mathbf{S}'(t)$ for memory mixing by extracting node states from TGNN temporal memory for nodes stored in the node bank. Expander embeddings $\mathbf{H}'(t)$ are computed over expander graph with adjacency matrix $\mathbf{A}^{\mathrm{Cay}}$ as:

$$\mathbf{H}'(t) = \mathrm{GNN}(\mathbf{S}'(t), \mathbf{A}^{\mathrm{Cay}}).$$

The input feature vector $\mathbf{X}(t)$ for TGNN forward pass is constructed through a concatenation ($\|$) of expander embeddings $\mathbf{H}(t^-)$ (for previously observed nodes) and TGNN node states $\mathbf{S}(t^-)$ (for new nodes) from temporal memory, where $t^-$ denotes time stamp just before $t$:

$$\mathbf{X}(t) = \mathbf{H}(t^-) \| \mathbf{S}(t^-).$$

## 5 Experiments

We evaluate TGR framework on the dynamic link property prediction task leveraging publicly available data on Temporal Graph Benchmark (Huang et al., 2023, TGB). We demonstrate TGR improves performance of existing TGNN architectures by implementing TGR on top of state-of-the-art models (Rossi et al., 2020; Zhang et al., 2024) featured on TGB benchmark. We further highlight that TGR-TGN achieves state-of-the-art on `tgbl-review` and TGR-TNCN achieves state-of-the-art on `tgbl-coin`, `tgbl-comment` and `tgbl-flight` datasets as of the time of writing.

**Datasets**. We leverage availability of temporal graph datasets made open-source through TGB Benchmark to validate TGR performance. TGB Benchmark collects a variety of real-world temporal networks to provide large-scale datasets spanning across a variety of domains such as flights, transactions and beyond. We provide description of each dataset in Appendix B.1.

**Baselines**. We demonstrate TGR performance by implementing TGR on top of the TGN (Rossi et al., 2020) and TNCN (Zhang et al., 2024) base models and study their performance on TGB temporal link prediction task (Huang et al., 2023). **TGN** (Rossi et al., 2020) is characterised by use of TGN temporal memory to store and update node states with use of recurrent neural networks such as LSTM (Hochreiter & Schmidhuber, 1997) or GRU (Cho et al., 2014). **TNCN** (Zhang et al., 2024) achieves state-of-the-art results on TGB link prediction task on a majority of datasets, by encoding link embeddings using neural common neighbour (Wang et al., 2024) in temporal setting.

Table 1: MRR for *dynamic link property prediction* on TGB Benchmark with baseline results imported directly from the leaderboard. First, second and third best performance are marked in **red**, **blue** and **bold** respectively. '–' means the method was omitted due to OOM (Huang et al., 2023).

| Model | tgbl-wiki | | tgbl-review | | tgbl-coin | | tgbl-comment | | tgbl-flight | |
|---|---|---|---|---|---|---|---|---|---|---|
| | Val | Test | Val | Test | Val | Test | Val | Test | Val | Test |
| **TGR-TNCN** | **75.1** | **72.4** | **51.1** | **59.9** | **76.9** | **78.3** | **89.9** | **89.1** | **85.4** | **85.1** |
| TNCN | 74.1 | 71.8 | 32.5 | 37.7 | **74.0** | **76.2** | **64.3** | **69.7** | **83.1** | **82.0** |
| Improvement (%) | 1.0 | 0.6 | 18.6 | 22.2 | 2.9 | 2.1 | 25.6 | 19.4 | 2.3 | 3.1 |
| **TGR-TGN** | 64.2 | 58.9 | **83.4** | **85.4** | 73.4 | 75.5 | 72.8 | 72.9 | 77.0 | 76.2 |
| TGN | 43.5 | 39.6 | 31.3 | 34.9 | 60.7 | 58.6 | 35.6 | 37.9 | 73.1 | 70.5 |
| Improvement (%) | 20.7 | 19.3 | 52.1 | 50.5 | 12.7 | 16.9 | 37.2 | 35.0 | 3.9 | 5.7 |
| DyRep | 7.2 | 5.0 | 21.6 | 22.0 | 51.2 | 45.2 | 29.1 | 28.9 | 57.3 | 55.6 |
| EdgeBank$_{tw}$ | 60.0 | 57.1 | 2.4 | 2.5 | 49.2 | 58.0 | 12.4 | 14.9 | 36.3 | 38.7 |
| EdgeBank$_{\infty}$ | 52.7 | 49.5 | 2.3 | 2.3 | 31.5 | 35.9 | 10.9 | 12.9 | 16.6 | 16.7 |
| DyGFormer | **81.6** | **79.8** | 21.9 | 22.4 | 73.0 | 75.2 | 61.3 | 67.0 | – | – |
| GraphMixer | 11.3 | 11.8 | **42.8** | **52.1** | – | – | – | – | – | – |
| TGAT | 13.1 | 14.1 | 32.4 | 35.5 | – | – | – | – | – | – |
| NAT | **77.3** | **74.9** | 30.2 | 34.1 | – | – | – | – | – | – |
| CAWN | 74.3 | 71.1 | 20.0 | 19.3 | – | – | – | – | – | – |
| TCL | 19.8 | 20.7 | 19.9 | 19.3 | – | – | – | – | – | – |

**TGR Implementation**. TGR is implemented on top of TGNN by adding node bank to store observed node IDs and memory mixing module to compute expander embeddings. Empirically, expander embeddings can enhance input node feature for observed nodes in the batch or additionally with their 1-hop neighbourhood. For experiments, we set this choice as a hyperparameter and select the best performing one based on validation set for each dataset. The experiements were conducted using either V100 or A100 GPUs. In all experiments we set learning rate $lr = 10^{-4}$ and run models with a tolerance that varies based on the size of dataset and computational time. We observe that added computational cost of rewiring in TGR is in most cases lower than the computational cost of base TGNN model. We maintain same batch size as in TGB and additionally provide a full list of hyperparameters in Appendix D.

**Results and Discussions**. We show that TGR significantly outperforms the base TGNN on the temporal link prediction task with largest improvement achieved across tgbl-review (50.5%), tgbl-comment (35.0%) and tgbl-wiki (19.3%) datasets. Full table of results is given in Table 1. We observe that TGR shows higher improvement across datasets with high surprise indices such are tgbl-review and tgbl-comment. Moreover, we identify that TGR achieves significant improvement over bipartite datasets such are tgbl-wiki and tgbl-review. Both bipartite and high surprise index temporal graphs posses a layer of information that is inaccessible to traditional TGNN architectures. Due to a nature of their structure, bipartite graphs lack connections between nodes of the same class. High surprise index datasets contain edges between nodes that do not have *obvious* connections in the training data, thus making it challenging for traditional TGNNs to distinguish. Both situations link to temporal under-reaching, previously discussed in Section 3.3, showing that TGR is best used to optimize input graphs that have high presence of temporal under-reaching. Consistent improvement across all datasets signals that TGR uncovers structural information which was previously inaccessible to TGNN base model.

## 6 CONCLUSION

In this work, we propose Temporal Graph Rewiring (TGR), a first approach for graph rewiring on temporal graphs to address limitations in TGNNs namely temporal under-reaching, over-squashing and memory staleness. We show that using expander graphs to rewire temporal graphs is an optimal method to address identified issues. This paper provides a promising first step towards applying temporal graph rewiring to optimize performance of TGNN models, paving the way for future temporal graph rewiring methods. In future work, we hope to further address identified issues by investigating other graph rewiring methods and compare their performance to TGR.

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

# A  OVERVIEW OF TGNN BASELINES

**TGN**. Temporal Graph Network (TGN) (Rossi et al., 2020) proposes a combination of temporal memory to store node states and graph-based operators for learning on continuous-time dynamic graphs. Following interaction in a CTDG, node states of the nodes involved in the event are updated in the memory through a recurrent neural network such as LSTM (Hochreiter & Schmidhuber, 1997) or GRU (Cho et al., 2014).

**TNCN**. TNCN builds up on memory-based TGNNs (Rossi et al., 2020; Trivedi et al., 2019) by introducing temporal version of a neural-common-neighbour (NCN) and reaching state-of-the-art results on three out of five datasets at TGB Benchmark.

# B  TGB BENCHMARK

This section provides overview and statistics of TGB Benchmark datasets (Huang et al., 2023).

## B.1  DATASET DESCRIPTION

**tgbl-wiki**. `tgbl-wiki` dataset stores dynamic information about a co-editing network on Wikipedia pages over a span of one month. The data is stored in a **bipartite** temporal graph where nodes represent editors or wiki-pages they interact with. An edge represents an action user takes when editing a Wikipedia page and edge features contain textual information about a page of interest. The goal is to predict existence and nature of links between editors and Wikipedia pages at a future timestamp.

**tgbl-review**. `tgbl-review` dataset stores dynamic information about Amazon electronics product review network covering the years 1997 to 2018. This dataset forms a **bipartite** weighted network, where nodes represent users and products, and edges signify individual reviews—rated on a scale from one to five—submitted by users to products at specific times. The goal is to predict which user will interact in the reviewing process at a given time.

**tgbl-coin**. `tgbl-coin` dataset stores cryptocurrency transactions extracted from the Stablecoin ERC20 transactions dataset. Nodes represent addresses, while edges indicate the transfer of funds between these addresses over a period of time. Covering the period from April 1 to November 1, 2022, the network includes transaction data for five stablecoins and one wrapped token. The goal of the task is to predict with which destination a given address will interact at a given time.

**tgbl-comment**. `tgbl-comment` dataset captures a directed network of Reddit user replies spanning from 2005 to 2010. Nodes represent individual users, and directed edges correspond to replies from one user to another within discussion threads. The goal of the task is to predict whether a pair of users will interact at a given time.

**tgbl-flight**. `tgbl-flight` dataset represents an international flight network from 2019 to 2022. Airports are modeled as nodes, and flights occurring between them on specific days form the edges. Node features include the airport type, the continent of location, ISO region codes, and geographic coordinates (longitude and latitude). Edge attributes consist of the associated flight numbers. The task is to predict if a given flight will exist between a source and destination airport at a given day.

## B.2  DATASET STATISTICS

Table 2: TGB Dataset characteristics studied in this work.

| Scale | Name | #Nodes | #Edges* | #Steps | Surprise | Metric |
|--------|--------------|---------|------------|------------|----------|--------|
| small | tgbl-wiki | 9,227 | 157,474 | 152,757 | 0.108 | MRR |
| small | tgbl-review | 352,637 | 4,873,540 | 6,865 | 0.987 | MRR |
| medium | tgbl-coin | 638,486 | 22,809,486 | 1,295,720 | 0.120 | MRR |
| large | tgbl-comment | 994,790 | 44,314,507 | 30,998,030 | 0.823 | MRR |
| large | tgbl-flight | 18,143 | 67,169,570 | 1,385 | 0.024 | MRR |

Table 2 shows the statistics of datasets used in this work from TGB. As shown, a wide range of datasets across multiple domains and scales are tested.

## C  ABLATION STUDY

**Choice of expander layer**. We compare TGR performance using *three* different baselines for expander layer: *TGR-GCN* (Kipf & Welling, 2016), *TGR-GAT* (Veličković et al., 2017) and *TGR-GIN* (Xu et al., 2018). Results of the study show that the effect of expander layer choice varies but that performance is largely independent of the choice of layer. For experiments, we use the GAT layer which has the highest empirical performance on `tgbl-review`.

Table 3: Performance comparison of expander layers on `tgbl-wiki` and `tgbl-review` datasets.

| Model | tgbl-wiki | | tgbl-review | |
|---|---|---|---|---|
| | Val MRR | Test MRR | Val MRR | Test MRR |
| TGR-TGN-GAT | 64.2 | 58.9 | 83.4 | 85.6 |
| TGR-TGN-GCN | 64.0 | 59.6 | 82.9 | 85.2 |
| TGR-TGN-GIN | 64.8 | 60.1 | 78.3 | 81.7 |

## D  MODEL PARAMETERS

We report TGR hyperparameters in Table 4.

Table 4: Model Hyperparameters.

| | Value |
|---|---|
| Temporal Memory Dimension | 100 |
| Node Embedding Dimension | 100 |
| Time Embedding Dimension | 100 |
| Expander Memory Dimension | 100 |
| Expander Embedding Dimension | 100 |
| # Attention Heads | 2 |
| Dropout | 0.1 |

