# OpenReview forum: "Temporal Graph Rewiring with Expander Graphs"
_ICLR.cc/2025/Conference — ICLR 2025 Conference Withdrawn Submission_

### Official Review · Reviewer_aU25 · 2024-10-25

**Soundness:** 2
**Presentation:** 4
**Contribution:** 2
**Rating:** 5
**Confidence:** 4

**Summary:**

This paper focuses on temporal graph learning and proposes a Temporal Graph Rewiring (TGR) method for graph rewiring on temporal graphs. TGR uses expander graph propagation to construct message-passing highways between graph nodes for rewiring. The experimental results on temporal graph benchmarks (TGB) demonstrate the effectiveness and scalability of the proposed TGR compared to baselines.

**Strengths:**

+ The writing is clear, and the paper is well-organized.
+ The approach is interesting and addresses an important topic.
+ Code for reproducing the experiments is provided.

**Weaknesses:**

+ The proposed methodology is limited to continuous-time temporal graphs, and further work would be needed to extend the model to discrete-time settings.
+ The experiments are conducted on a temporal link prediction task, which is insufficient to fully validate the performance of TGR. It is suggested to include node-level tasks, such as dynamic node classification, to demonstrate the broader applicability of TGR.
+ Although TGR is proposed to address under-reaching and over-squashing, there are no empirical results to support these claims.

**Questions:**

+ Can the proposed TGR process newly arrived nodes/edges in an inductive manner?
+ What is the time and memory complexity of TGR? Are there any empirical results?
+ Why are static graph rewiring methods mentioned in related work but missing from the comparison?
+ If I understand correctly, the rewiring is achieved by defining new message-passing connections rather than actually modifying the graph topology?

---

### Official Review · Reviewer_zfq4 · 2024-11-01

**Soundness:** 2
**Presentation:** 1
**Contribution:** 1
**Rating:** 3
**Confidence:** 4

**Summary:**

The paper identifies the under-reaching effect in temporal graphs and argues that temporal graphs may suffer from a more severe under reaching effect than that of static graphs. The paper then proposes to use the expander graph propagation (EGP) algorithm to alleviate the issue, via generalizing the EGP method to temporal graphs. Empirically, the resulting framework, TGR, achieves state-of-the-art performance over the TGB benchmark using standard temporal GNN backbones.

**Strengths:**

- The under-reaching and over-squashing phenomenon over temporal graphs is a nascent research field. The paper provide insights that such issues might be overlooked in previous studies on temporal graph representation learning.
- The proposed TGR framework can serve as a backbone-agnostic augmentation to any off-the-shelf temporal graph neural models, which is flexible with strong empirical performance.

**Weaknesses:**

- Limited novelty: The main algorithmic contribution of the paper, TGR, is an ad-hoc combination of the EGP method [1] with standard temporal GNN protocols.
- Lack of empirical evidence to under-reaching in temporal graphs: In section 3.3 of the paper, the authors state that the under-reaching effect in temporal graphs might be more severe than that in static graphs. While the authors present two propositions, the constructions therein are contrived and not realistic. It is desirable that the authors present more examples from real-world temporal graphs that illustrates the under-reaching phenomenon.

**Questions:**

Asides from those raised in the weakness section, I have the following questions:
- While it was stated in the paper that the primary focus is temporal link prediction, I am curious about how TGR performs regarding node-level tasks like dynamic node property prediction.
- In section 4.1 (line 402-405), the authors describe the expander graph construction as ``a large expander graph is pre-computed at the beginning of training``, how to determine the size of the expander graph?



[1]. Deac, Andreea, Marc Lackenby, and Petar Veličković. "Expander graph propagation." Learning on Graphs Conference. PMLR, 2022.

---

### Official Review · Reviewer_8MCS · 2024-11-01

**Soundness:** 3
**Presentation:** 2
**Contribution:** 3
**Rating:** 5
**Confidence:** 4

**Summary:**

The authors discuss a variant of the under-reaching phenomenon in memory-based Temporal Graph Networks (TGNs) and propose a solution to alleviate this.

They first define the problem of under-reaching in continuous time dynamic graphs and then discuss how this can occur in an abstract setting for TGNs, highlighting the fact that this phenomenon may generally be more frequent than in static graphs and standard Graph Neural Networks.

Successively, the authors move to discuss differences and additional complications arising from practical implementation and training details.

The discussed phenomenon is addressed by the use of a memory mixing component that alleviates under-reaching by performing message-passing on an expander graph constructed on the set of observed nodes. The resulting approach is dubbed TGR.

Experimental results show how TGR significantly enhances the performance of base neural temporal architectures. The authors underscore the fact that these enhancements are more evident in the case of bipartite dynamic graphs and settings with high-surprise rates.

**Strengths:**

- The paper sheds light on an interesting and seemingly relevant phenomenon; the community may likely benefit from the explicit formalisation of this problem as the authors set to do in this manuscript;
- The discussions generalising the analysis to consider practical implementation and training details is interesting and useful;
- The approach proposed to alleviate the highlighted problem is simple and effective, other than already known and studied by the community in other contexts.

**Weaknesses:**

- In the current form, the manuscript is not optimally presented and is not making effective use of the space available in the main paper. In particular, the presentation could be largely improved by:
    - Adding more exemplary illustrations to visually support the analyses and results on the temporal under-reaching phenomenon;
    - Relegating to appendix subsections illustrating concepts related to static graphs; at the moment they are seemingly presented in more detail than those on dynamic graphs (see e.g. 4.1, 4.2);
    - Adding a more detailed (mathematical) description of the approach in Section 4.3, which appears to be too shallow: e.g. the exact steps involved in the memory update are not clear.
- The core Sections 3.3, 3.4 are quite interesting, but, at the moment, they read high-level. In particular, the reviewer believes:
    - Propositions 1 and 2 would require a more precise formalisation and proof (that could potentially be sketched and deferred in full to the appendix);
    - Exemplary cases leveraged to convey the arguments in both the two sections should be illustrated and clearly referenced to ease the comprehension of the readers: they could, e.g., explicitly provide some under-reaching configurations on path- and non-path dynamic graphs to convey the meaningfulness of Definition 2, or the impact of batching.
- It is not clear — or otherwise not discussed — whether the described phenomenon occurs and is relevant in the case the considered dynamic graphs do not possess node features.
    - Which of the datasets considered in the experimental section contain node features? To what we could reasonably attribute the performance enhancements of TGR in the case they do not?
- The under-reaching analysis is developed specifically to the content of the memory module. The discussion is extended to temporal node embeddings in Section 3.5, but a more systematic treatment would be expected.
    - To the best of reviewer’s understanding, message-passing of the memory content on the graph induced by past interactions could, in fact, alleviate the under-reaching phenomenon. However, the authors state that considering two-hop neighbourhoods “only reduces our path constraint for temporal mixing by two steps” which seems to instead convey the problem is *not* exacerbated. The authors should be more clear in discussing this aspect.
    - Can the authors more formally generalise Propositions 1 and 2 to also consider a fixed number of propagations to obtain temporal embeddings?
- The authors hypothesise that memory mixing is beneficial in the case of bipartite graphs and settings of high surprise rate, as they claim “[…] TGR uncovers structural information which was previously inaccessible to TGNN base model”. This claim is strong while being not formally backed.
    - The paper would benefit from further validating this hypothesis in a controlled setting, e.g. a synthetic benchmark.
    - Can the authors comment on this aspect, either way?

**Questions:**

- Have the authors experimented with other simple approaches to the proposed problem? For example, can the use of a “Virtual Node” — an artificial node connected to all others to induce global information propagation — be equally beneficial? In general there may be other simple ways to approach the under-reaching problem, but the authors do not seem to discuss them in comparison to the use of expander graphs.
- Can the authors be more clear on what they mean by a concatenation of expander embeddings and TGNN node states? On which axis does it take place?
- It is not clear to the reviewer if the presented phenomenon is best understood and presented as the temporal counterpart of under-reaching on static graphs, or rather, as a problem due to updating the memory module according to a causal scheme. After all, message-passing of the memory content performed to obtain temporal embeddings could allow to “reach” nodes and counteract this phenomenon. Can the authors comment on this?
- Can the authors expand on how the memory is updated, when, and based on what kind of information? From Figure 2 it seems that the memory component would not receive gradients during training (an aspect discussed and approached in the original TGN paper).

Also, see above “Weaknesses”.

---

### Official Review · Reviewer_Q24G · 2024-11-03

**Soundness:** 3
**Presentation:** 3
**Contribution:** 2
**Rating:** 5
**Confidence:** 3

**Summary:**

This paper introduces Temporal Graph Rewiring (TGR), a novel method proposed to address the dominant vulnerabilities of TGNNs regarding under-reaching and over-squashing issues, which are often exacerbated in temporal graphs due to stale nodes and edges. This paper provides theoretical motivation on this front. Previously, graph rewiring techniques were applied only to static graphs; this paper is the first to apply these techniques to temporal graphs. It uses expander graph propagation to create efficient message-passing pathways between temporally distant nodes. The designed component shows significant performance improvement over the existing state-of-the-art benchmarks on the TGB benchmark, which includes large-scale and diverse datasets.

**Strengths:**

- This paper provides solid theoretical motivation concerning how under-reaching and over-squashing issues are exacerbated in the context of dynamic graph problems.

- The experiments are thorough on the Temporal Graph Benchmark (TGB). The datasets in TGB appear to be large-scale, providing sufficient experimental justification for the proposed techniques.

- The overall presentation of the work is good, and the designed method is easy to understand.

**Weaknesses:**

- The paper is directly built on the existing work of Deac et al. (2022), extending it to a new dynamic graph setting. This integration appears straightforward as the use of expander graphs is not specifically tailored for temporal graphs.

- The authors seem to believe that the only way to address the issues of under-reaching and over-squashing is through graph rewiring using expander graphs. However, under-reaching and over-squashing are commonly acknowledged problems in graph theory, and there are other ways to address these issues, such as virtual node design, changing model architectures, and altering the temporal positional-encoding design, etc. This paper lacks a discussion on alternative solutions to this issue. It seems as though the paper is trying to use the "expander graph" as a hammer to find a nail - the "under-reaching issue in dynamic graphs" - rather than a more natural, problem-driven research approach. Please refer to the following literature, which all address some aspect of the similar issue: [1], [2], [3], [4].

- In the background and motivation section, more discussion on the use of expander graphs for static graphs could be included. Since this is the key prior knowledge that the paper relies on, it would help provide more background information to the readers.

[1] On the Connection Between MPNN and Graph Transformer. ICML 2023.
[2] An Analysis of Virtual Nodes in Graph Neural Networks for Link Prediction. LoG 2022.
[3] Understanding Virtual Nodes: Oversmoothing, Oversquashing, and Node Heterogeneity. Arvix 2405.13526, 2024.
[4] Todyformer: Towards Holistic Dynamic Graph Transformers with Structure-Aware Tokenization. TMLR 24.

**Questions:**

- Please refer to the comments above regarding limitations and address the issue mentioned in the limitations.

- In Figure 2, why is the decoder not placed after the output of the Expander GNN H′(t), but instead after the TGNN output Z(t)?

**Details Of Ethics Concerns:**

This paper doesn't seem to have ethic concern for me.

---

### Note · Authors · 2024-12-03

**Comment:**

Dear Reviewers,

We thank you all for taking the time taken to go through our work and provide valuable comments. After assessing your feedback, we have decided to withdraw our submission at this time and re-work the paper to respond to the points that have been highlighted which require time beyond the rebuttal period. We highly appreciate the questions that have been raised during this rebuttal period and we intend to incorporate changes ahead of the re-submission to the next appropriate venue.

Yours sincerely,
Authors of TGR

**Withdrawal Confirmation:**

I have read and agree with the venue's withdrawal policy on behalf of myself and my co-authors.